# Metabolomic Profiling Reveals Protective Effects and Mechanisms of Sea Buckthorn Sterol against Carbon Tetrachloride-Induced Acute Liver Injury in Rats

**DOI:** 10.3390/molecules27072224

**Published:** 2022-03-29

**Authors:** Changting Sheng, Yang Guo, Jing Ma, Eun-Kyung Hong, Benyin Zhang, Yongjing Yang, Xiaofeng Zhang, Dejun Zhang

**Affiliations:** 1College of Medicine, Qinghai University, Xining 810016, China; changting.sheng01@gmail.com (C.S.); 1910020117@qhu.edu.cn (Y.G.); 2College of Ecological and Environmental Engineering, Qinghai University, Xining 810016, China; 190901j1206@qhu.edu.cn (J.M.); byzhang@nwipb.cas.cn (B.Z.); yongjing223@163.com (Y.Y.); 2021990035@qhu.edu.cn (X.Z.); 3Medvill Co., Ltd., Medvill Research Institute, Seoul 100744, Korea; ekhongmed@medvill.co.kr

**Keywords:** acute liver injury, metabolomics, carbon tetrachloride, sea buckthorn sterol

## Abstract

The present study was designed to examine the efficacy and protection mechanisms of sea buckthorn sterol (SBS) against acute liver injury induced by carbon tetrachloride (CCl_4_) in rats. Five-week-old male Sprague-Dawley (SD) rats were divided into six groups and fed with saline (Group BG), 50% CCl_4_ (Group MG), or bifendate 200 mg/kg (Group DDB), or treated with low-dose (Group LD), medium-dose (Group MD), or high-dose (Group HD) SBS. This study, for the first time, observed the protection of SBS against CCl_4_-induced liver injury in rats and its underlying mechanisms. Investigation of enzyme activities showed that SBS-fed rats exhibited a significant alleviation of inflammatory lesions, as evidenced by the decrease in cyclooxygenase-2 (COX-2), prostaglandin E2 (PGE2), and gamma-glutamyl transpeptidase (γ-GT). In addition, compared to the MG group, the increased indices (superoxide dismutase (SOD), glutathione peroxidase (GSH-Px), catalase (CAT), total antioxidant capacity (T-AOC), and total protein (TP)) of lipid peroxidation and decreased malondialdehyde (MDA) in liver tissues of SBS-treated groups showed the anti-lipid peroxidation effects of SBS. Using the wide range of targeted technologies and a combination of means (UPLC-MS/MS detection platform, self-built database, and multivariate statistical analysis), the addition of SBS was found to restore the expression of metabolic pathways (e.g., L-malic acid, N-acetyl-aspartic acid, N-acetyl-l-alanine, etc.) in rats, which means that the metabolic damage induced by CCl_4_ was alleviated. Furthermore, transcriptomics was employed to analyze and compare gene expression levels of different groups. It showed that the expressions of genes (Cyp1a1, Noct, and TUBB6) related to liver injury were regulated by SBS. In conclusion, SBS exhibited protective effects against CCl_4_-induced liver injury in rats. The liver protection mechanism of SBS is probably related to the regulation of metabolic disorders, anti-lipid peroxidation, and inhibition of the inflammatory response.

## 1. Introduction

Sea buckthorn is a kind of small tree or deciduous shrub in the Elaeagnaceae family [1]. Previous studies have shown that sea buckthorn contains many types of chemical components and rich bioactive substances, offering more than 190 active ingredients [2]. In addition, 103 active ingredients have been found in sea buckthorn fruit oil [3]. Sea buckthorn also has anti-inflammatory, anti-cancer, anti-aging, immunological modification, and anti-oxidation effects, as well as the inhibition of cholesterol absorption and the reducing of blood lipids [4]. The wide usage of sea buckthorn in healthy food, wine, and natural medicine makes it a natural product with ecological, social, and economic benefits [5]. 

The liver is an important organ for protein synthesis, degradation, and detoxification, as well as amino acid metabolism. Nevertheless, the liver is easily damaged by various exogenous materials, causing acute or chronic liver diseases [6]. Therefore, it is desired to study the liver protection mechanism. 

CCl_4_ is commonly used in a mouse model of liver injury, which is responsible for oxidative stress and lipid peroxidation through the cytochrome P450-mediated generation of the highly reactive CCl_3_, leading to eventual cellular damage characterized by hepatocellular necrosis [7]. The biochemical and histological events of corresponding damage induced by CCl_4_ are quite similar to liver cirrhosis in humans. Through oral or gavage feedings, reproducible and obvious acute liver injury can be induced, helping in the study of pathogenic mechanisms or protection mechanisms related to the liver. β-Sitosterol, a naturally occurring sterol molecule, is a relatively mild to moderate antioxidant and exerts beneficial effects in vitro by decreasing the level of reactive oxygen species. A previous study [8] evaluated the antioxidant potential of β-Sitosterol in 1,2-dimethylhydrazine (DMH)-induced colon carcinogenesis. There was another study that suggested that phytosterol esters have [9] effects on countering hypercholesterolemia-related changes in the brain by decreasing the cholesterol levels, increasing the phospholipid levels, and increasing the level of antioxidant enzymes. 

Extensive studies have shown that sea buckthorn fermentation liquid protects against alcoholic liver disease. The underlying protection mechanism may involve regulations of liver lipid metabolism and oxidative stress [10]. Dietary supplements that contain sea buckthorn for hyperlipidemia treatment are useful in minimizing the oxidative damage caused by lipid peroxidation [11]. In addition, a previous work [12] indicated that pretreatment with sea buckthorn protects against LPS/d-GalN-induced liver injury in mice by suppressing the TLR4-NF-κB signaling pathway, which means that sea buckthorn may be a promising drug for the prevention of acute live injury. Furthermore, sea buckthorn inhibits CYP3A down-regulation during immune liver injury, primarily through the transcriptional regulation and post-translational nitration of enzyme proteins. This mechanism is also applicable to alcoholic liver injury and non-alcoholic fatty liver [13]. On the other hand, many investigations demonstrate that SO_2_ exposure causes a significant change in the glutathione redox system of rats. This change might be one of the possible mechanisms by which SO_2_ inhalation leads to lipid peroxidation. Sea buckthorn seed oil was found to be protective against oxidative damage induced by SO_2_ [14]. Furthermore, sea buckthorn berry juice can alleviate symptoms of oxidative stress and tissue damage by preventing the destruction of antioxidant enzymes and inhibiting lipid peroxidation [15]. The extract of Hippophae rhamnoides as well as vitamin E can protect the liver against nicotine-induced oxidative stress [16].

The studies mentioned above have shown that sea buckthorn has multiple functions, especially in liver protection and anti-oxidation. However, the existing studies focus mainly on sea buckthorn oil, sea buckthorn berry juice, and sea buckthorn fermentation liquid, which are crude extracts of sea buckthorn. There are surprisingly few studies investigating the liver protection mechanism of specific components of sea buckthorn. In addition, previous related works barely use different types of newer techniques, such as transcriptomics and metabonomics, to comprehensively analyze the action mechanisms of sea buckthorn in CCl_4_-induced liver injury rats.

In the present study, we focus on analyzing the liver protection mechanism of sterols extracted from sea buckthorn for the first time. In this study, sea buckthorn sterol pre-treatment was used. The pretreatment experiment is a common way to study the protective effect of a substance that needs to be analyzed [17]. Additionally, a variety of research methods (e.g., transcriptomics and metabonomics) were employed, which made it possible for us to carefully and accurately explain the protection mechanisms of sea buckthorn. In our previous studies, we reported that sea buckthorn regulates inflammatory factors and transaminases in the blood of CCl_4_-induced liver injury rats [18]. The present work further demonstrates the protective effects of sea buckthorn sterol (SBS) on the liver, and the corresponding protection mechanisms are comprehensively analyzed from multiple aspects. 

## 2. Materials and Methods

### 2.1. Animals, Reagents, and Drugs

Sixty five-week-old male Sprague-Dawley (SD) rats, of SPF grade, provided by Chongqing, China, Tengxin Biotechnology Co., Ltd. (License: SCXK[Shan] 2018-001) were used in this study. All animal studies were approved by the animal ethics committee of Qinghai University. The rats anesthetized by intraperitoneal injection of 20% urethane (0.5 mL/100 g) and then euthanized by bloodletting of the celiac artery (Audit Form No.: 2019-1). The SBS was extracted in our laboratory (seed oil of sea-buckthorn was purchased from Qinghai, China, Qinghua Bozhong Biotechnology Co., Ltd.) by the following processes: (i) extraction crude sterol from seabuckthorn seed oil by saponification; (ii) repeat recrystallization with anhydrous ethanol until SBS was purified. SBS has no hepatotoxicity, which has been validated in this study and other studies [19]. CCl_4_ (batch number: 56-23-5) and Bifendate (batch number: D25O10G100857) were purchased from Qinghai, China, Rhine Biotechnology Co., Ltd. The cyclooxygenase-2 (COX-2) kit (batch number: uifhwf41eb) and prostaglandin E2 (PGE2) kit (batch number: kex55l2533) were purchased from Elabscience Biotechnology Co., Ltd. (Wuhan, China). The superoxide dismutase (SOD) detection kit (batch number: 20191123), glutathione peroxidase (GSH-PX) kit (batch number: 20191118), gamma-glutamyl transpeptidase (γ–GT) kit (batch number: 20191112), catalase (CAT) kit (batch number: 20191120), total antioxidant capacity (T-AOC) kit (batch number: 20191116), total protein (TP) kit (batch number: 20191115), and malondialdehyde (MDA) kit (batch number: 20191119) were purchased from Nanjing Jiancheng Bioengineering Institute. Methanol, acetonitrile, ethanol, acetic acid, ammonium methyl acetate, chloroform, and methyl tert-butyl ether were supplied by Merck at HPLC grade, and the standard materials were supplied by BioBioPha. The standard materials were dissolved in dimethyl sulfoxide (DMSO) or methanol, and were stored at −20 °C. They were diluted with 70% methanol before mass spectrometry in gradient concentrations.

### 2.2. Preparations of the Acute Liver Injury Model 

Sixty male SD rats were randomly divided into six groups (10 rats in each group): a blank control group (BG, oral administration of saline), a model group (MG, oral administration of saline), a bifendate control group (DDB, oral administration of bifendate 200 mg/kg), and low-dose (LD), medium-dose (MD), and high-dose (HD) SBS-treated groups (oral administration of 100, 200, 400 SBS mg/kg, respectively). All materials were treated for 7 days. After 2 h from the last oral administration, 50% CCl_4_ solved in olive oil (2 mL/kg) were administered orally to all animal groups except BG. The BG group was given the same volume of saline orally.

### 2.3. Pathological Examination of Liver Tissue 

Three rats in each group were randomly selected and anesthetized by intraperitoneal injection of 20% urethane. After laparotomy, samples were collected from the edge of the left lobe of the liver and placed in 3% dialdehyde. Ultrathin sections were obtained through the processes of fixing, dehydrating, and permeation embedding. Before observing these sections using the transmission electron microscope, we used the uranium acetate and lead citrate to stain them successively. In the same three rats, samples collected at the edge of the right lobe of the liver were placed and fixed in 10% neutral formaldehyde. After dehydration, pruning, embedding, sectioning, staining, sealing, and HE staining were performed, the histopathological changes of the rats’ livers in each group were observed under a light microscope.

### 2.4. Measurement of Biochemical Index in Liver Tissue Homogenate

Liver tissues were mixed with saline at a ratio of 1:9, and then the mixture was homogenized to make the liver tissue homogenate. After removing the supernatant, the levels of SOD, GSH-Px, Cat, γ-GT, T-AOC, total protein, MDA, COX-2, and PGE_2_ were measured in the liver tissue homogenate according to the instructions of the test kit.

### 2.5. Sample Preparation

#### 2.5.1. Extraction of Hydrophilic Compounds

A quantity of 50 mg of sample was taken and homogenized with 1 mL of ice-cold methanol/water (70%, *v*/*v*). Cold steel balls were added to the mixture and homogenized at 30 Hz for 3 min. After stirring the mixture for 1 min, it was centrifuged in 12,000 rpm at 4 °C for 10 min. The collected supernatant was later used for LC-MS/MS analysis.

#### 2.5.2. Extraction of Hydrophobic Compounds

A quantity of 50 mg of sample was taken and homogenized with 1 mL of mixture (including methanol, MTBE, and internal standard mixture) and cold steel balls were added. After the steel balls were removed, the mixture was stirred for 2 min. Subsequently, 500 μL of water was added and the mixture was stirred for 1 min, and then it was centrifuged in 12,000 rpm at 4 °C for 10 min. Following this, 500 μL of supernatant was concentrated and dried. The powder was dissolved in 100 μL of mobile phase B, and stored at −80 °C. Finally, the dissolving solution was taken into the sample bottle for LC-MS/MS analysis.

### 2.6. Analysis by UPLC

The samples were analyzed using the LC-ESI-MS/MS system (UPLC, Shim-pack UFLC SHIMADZU CBM A system, https://www.shimadzu.com/ (accessed on 21 February 2020); MS, QTRAP^®^ System, https://sciex.com/ (accessed on 21 February 2020)).

The UPLC conditions of hydrophilic compounds were as follows. A Waters ACQUITY UPLC HSS T3 C18 (1.8 µm, 2.1 mm × 100 mm) column was used. The column temperature was 40 °C. The flow rate was 0.4 mL/min, and the injection volume was 2 μL. The mobile phase was water (0.04% acetic acid): acetonitrile (0.04% acetic acid). The gradient condition was as follows: 95:5 *v*/*v* at 0 min, 5:95 *v*/*v* at 11.0 min, 5:95 *v*/*v* at 12.0 min, 95:5 *v*/*v* at 12.1 min, 95:5 *v*/*v* at 14.0 min.

The UPLC conditions of hydrophobic compounds were as follows. A Thermo C30 (2.6 μm, 2.1 mm × 100 mm) column was used. The mobile phase was a mixed solvent gradient system with the following contents: A—acetonitrile/water (60/40 V, 0.04% acetic acid, 5 mmol/L ammonium formate); and B—acetonitrile/isopropanol (10/90 V, 0.04% acetic acid, 5 mmol/L ammonium formate). The gradient condition was as follows: A/B (80:20 *v*/*v*) at 0 min, 50:50 *v*/*v* at 3.0 min, 35:65 *v*/*v* at 5 min, 25:75 *v*/*v* at 9 min, 10:90 *v*/*v* at 15.5 min. The flow rate was 0.35 mL/min. The column temperature was 45 °C, and the injection volume was 2 μL. The effluent was alternatively connected to an ESI-triple quadrupole-linear ion trap (QTRAP)-MS.

### 2.7. Analysis by ESI-Q TRAP-MS/MS

#### 2.7.1. ESI-Q TRAP-MS/MS of Hydrophilic Compounds

LIT and triple quadrupole (QQQ) scans were acquired on a triple quadrupole-linear ion trap mass spectrometer (QTRAP), QTRAP^®^ LC-MS/MS System, equipped with an ESI Turbo Ion-Spray interface, operating in positive and negative ion mode and controlled by Analyst 1.6.3 software (Sciex). The ESI source operation parameters were as follows: the source temperature was 500 °C, the ion spray voltage (IS) was 5500 V (positive) and −4500 V (negative), the ion source gas I (GSI), gas II (GSII), and curtain gas (CUR) were set at 55, 60, and 25.0 psi, respectively, and the collision gas (CAD) was high. Instrument tuning and mass calibration were performed with 10 and 100 μmol/L polypropylene glycol solutions in QQQ and LIT modes, respectively. A specific set of MRM transitions was monitored for each period according to the metabolites eluted within this period.

#### 2.7.2. ESI-Q TRAP-MS/MS of Hydrophobic Compounds

LIT and triple quadrupole (QQQ) scans were acquired on a triple quadrupole-linear ion trap mass spectrometer (Q TRAP), API QTRAP LC/MS/MS System, equipped with an ESI Turbo Ion-Spray interface, operating in a positive ion mode and controlled by Analyst 1.6 software (AB Sciex). The ESI source operation parameters were as follows: the ion source, turbo spray, and source temperature were 550 °C, the ion spray voltage (IS) was 5500 V, the ion source gas I (GSI), gas II(GSII), and curtain gas (CUR) were set at 55, 60, and 25.0 psi, respectively, and the collision gas (CAD) was high. Instrument tuning and mass calibration were performed with 10 and 100 μmol/L polypropylene glycol solutions in QQQ and LIT modes, respectively. QQQ scans were acquired as MRM experiments with collision gas (nitrogen) set to 5 psi. DP and CE for individual MRM transitions were conducted with further DP and CE optimization. A specific set of MRM transitions was monitored for each period according to the metabolites eluted within this period.

### 2.8. Transcriptome Analysis

#### 2.8.1. RNA Detection

There were three steps: (i) Agarose gel electrophoresis: the integrity of RNA was analyzed and it was determined whether there was DNA pollution; (ii) Qubit 2.0 fluorometer: high-precision measurement of RNA concentration was performed; (iii) Agilent 2100 biological analyzer: accurate detection of RNA integrity was performed.

#### 2.8.2. Library Setup

There are two ways to obtain mRNA: one is to enrich the mRNA with poly-A tail by oligo (DT) magnetic beads using most of the mRNA of eukaryotes with poly-A tails. The other is to remove ribosomal RNA from total RNA to obtain mRNA, and then add the fragmentation buffer. The first strand of cDNA was synthesized by random hexamers. The second strand cDNA was synthesized by adding buffer, dNTPs (dUTP, dATP, dGTP, and DCTP), and DNA polymerase I. The double-stranded cDNA was purified by AMPure XP beads. Then, end repair, addition of the A-tail, and the connecting of sequencing adaptors were performed on the purified double-stranded cDNA. Next, AMPure XP beads were used for fragment size selection. Finally, the final cDNA library was obtained by performing PCR enrichment.

##### Library Quality Check

(i)Qubit2.0 was used for preliminary quantification, and Agilent 2100 was used to detect the insert size of the library. The next experiment could only be carried out after the insert size met expectations;(ii)The Q-PCR method accurately quantified the effective concentration of the library (the effective concentration of the library was >2 nM). Then, the library check was successfully performed.

##### Sequencing on the Machine

After the library was qualified, different libraries were pooled according to the target offline data volume and sequenced using the Illumina hiseq platform.

### 2.9. Statistical Analysis

SPSS 26.0 statistical software was used to analyze the data, expressed as mean ± standard deviation, and ANOVA was used for data analysis, followed by Bonferroni post hoc tests, with *p* < 0.05 determining statistical significance. After the LC-MS/MS analysis of metabolomic data, the mass spectrometry data were processed by Software Analyst 1.6.3. The genomic data filtered off-line data to obtain evident data. After performing comparisons with the designated reference genome, mapped data were obtained that were used for structural level analysis, such as alternative splicing analysis, new gene discovery, and gene structure optimization. According to the expression levels of genes in different samples or different groups, differential expression analysis, differential expression gene function annotation, function enrichment, and other expression levels were analyzed.

## 3. Results

### 3.1. Effects of SBS on SOD, MDA, and GSH-P_X_ in Rats with CCl_4_-Induced Liver Injury

As shown in Figure 1, compared with group BG, the levels of SOD and GSH-Px in the MG group decreased significantly, while the MDA content showed a considerable increase (*p* < 0.01). Compared to the MG group, the MD and HD groups exhibited an increase in the SOD level with statistically significant differences (*p* < 0.01). The level of SOD in the MD and HD groups was close to that in group BG. The values of MDA content in the LD, MD, and HD groups were all significantly decreased (*p* < 0.05 or *p* < 0.01), and the BG group showed a similar MDA content value to group HD. In addition, there was an increase in the level of GSH-Px in the LD, MD, and HD groups, with statistically significant differences (*p* < 0.01). Furthermore, the level of GSH-Px in the HD group was close to that in group BG. These results indicated that SBS has the effects of anti-lipid peroxidation and reducing lipid peroxidation products in the defense system of the body.

### 3.2. Effects of SBS on CAT and T-AOC in Rats with CCl_4_-Induced Liver Injury

Figure 2 demonstrates that the CAT and T-AOC levels in group MG were considerably lower than those in the BG (*p* < 0.01) group. Compared to the MG group, the levels of CAT in the LD, MD, and HD groups were significantly higher (*p* < 0.01). The increase in T-AOC level in the HD group showed a statistical significance (*p* < 0.01) compared with the MG group. These results suggested that SBS has the effect of anti-lipid peroxidation.

### 3.3. Effects of SBS on γ-GT, TP, PGE_2_, and COX-2 in Rats with CCl_4_-Induced Liver Injury

As shown in Figure 3 and Figure 4, the increased levels of γ-GT, PGE2, and COX-2, and the reduction in TP in the MG group were significant (*p* < 0.01) in comparison to the BG group. The level of γ-GT in HD was significantly lower than that in the MG group (*p* < 0.01). The comparison to group MG showed that the PGE_2_ levels in the MD and HD groups were significantly reduced (*p* < 0.01). The levels of PGE_2_ in the MD and HD groups were close to that in the BG group. Similar observations were also found by comparing the COX-2 levels in the SBS-treated group and the MG group, and a significant decrease (*p* < 0.01) was observed in the HD group. In addition, a close level of COX-2 in the HD group and BG group was found. As for the TP level, there was a considerable increase (*p* < 0.05 or *p* < 0.01) in the LD, MD, and HD groups, and two groups (MD and HD) shared a similar level of TP with the BG group. These results provided evidence that SBS can lower the level of inflammation and alleviate the liver injury.

### 3.4. Electron Microscopic Observations of Liver Tissue

An important observation could be obtained based on Figure 5, where the structure of liver tissues of group BG was obvious and normal. However, the MG group showed a fuzzy liver structure and obvious damage (e.g., necrotic liver cells, a large number of swollen mitochondria, many lipid droplets in the cytoplasm, and a large number of the expanded endoplasmic reticulum) was found. In the DDB group, the liver structure was partially clear, but some mitochondria were still slightly swollen and autophagic, and some liver cells were suspected to be necrotic. In addition, a small number of lipid droplets were formed. The liver structure of the LD and MD groups was clear, but there were still some rough expansions of endoplasmic reticulum and a small number of lipid droplets, as well as a few necrotic hepatocytes. The HD group had a relatively evident liver structure with abundant mitochondria in the cytoplasm, and the mitochondrial structure was basically obvious and clear. It was also found that necrotic hepatocytes and lipid droplets were greatly reduced.

These results of the electron microscopic observations showed that liver injury in the MG group was obvious in comparison to the BG group. The LD, MD, and HD groups showed different degrees of improvement in terms of liver damage expansion, among which the HD group showed the most obvious effect.

### 3.5. Light Microscopic Observations of Liver Tissue

Figure 6 and Table 1 show light microscopic observations of liver tissue. Glisson’s capsule of group BG was intact, and the morphology of hepatocytes was normal. Compared with group BG, fatty degeneration and feathery degeneration of hepatocytes were found, and the hepatocytes showed multiple punctate necrosis with obvious injury in the MG group. Comparisons between the MG and DDB groups indicated that Glisson’s capsule was intact and the degree of hepatocyte necrosis was significantly reduced in the DDB group. Additionally, Glisson’s capsule of two groups (LD and MD) was slightly damaged, and only a few hepatocytes were necrotic, as well as showing a small amount of fatty degeneration and feathery degeneration. In particular, the HD group showed an intact Glisson’s capsule, and the degree of liver cell lesion was greatly reduced.

Observations from light microscopy demonstrated that the degree of pathological change in the MG group was significantly aggravated compared to the BG group. Through further analysis, it was concluded that in comparison to group MG, the degrees of pathological change in the LD, MD, and HD groups were all alleviated, and the alleviation in group HD was the most obvious.

### 3.6. Identification and Analysis of Metabolites

#### 3.6.1. Discriminant Analysis of Orthogonal Partial Least Squares Method

Based on the self-built Metware Database (MWDB), qualitative analysis was performed according to the retention time (RT), ion-pair information, and secondary spectrum data. The hierarchical clustering map (Figure 7) and the Partial least squares discriminant analysis (PLS-DA) shown in Figure 8 were used to analyze the metabolic pattern of each group. It was found that the global metabolic statuses were significantly different between group MG and group BG, indicating that CCl4 caused an obvious metabolic disorder in the liver. In addition, the global metabolic statuses of the LD, MD, and HD groups were highly different from that of the MG group. Meanwhile, the metabolic trend of the HD group was close to that of group BG. These results indicated that SBS can effectively regulate the metabolic disorders induced by CCl4. 

#### 3.6.2. Determination of Relevant Metabolite

The S-plot (Figure 8) and variable importance in projection (VIP) were used to identify potential metabolites. The altered metabolites with statistical significance were screened using S-plot, and they were located in the upper right quadrant or in the lower left quadrant (i.e., situated away from the origin). The metabolites that changed significantly were selected according to the fold change (more than 1.5 or less than 0.67), the VIP score (more than 1.0), and FDR < 0.05. The *t*-test and ANOVA were employed to test the significant difference of metabolites between the groups. Referring to literature reports and searching in the online database (HMDB), 17 metabolites (shown in Table 2) related to the inhibitory effect of SBS on CCl_4_-induced liver damage were selected. The criteria for selecting the metabolite was based on the selection of the common differential metabolites that appeared in each group, which were related to acute liver injury. In the MG group, the levels of L-malic acid, 7Z, 10Z, 13Z, 16Z, 19Z-docosapentaenoic acid, creatine, n-acetyl-l-alanine, N-acetylaspartate, Trigonelline, 4-guanidinobutyric acid, N-amidino-L-aspartate, CE (16:1), CE(18:2), PE (16:1/16:0), DG (16:0/20:2/0:0), TG (14:0/18:0/18:2), TG (14:0/18:0/20:4), and TG (16:0/16:1/22:5) were significantly increased, and levels of n-glycyl-l-leucine and FFA (6:0) were obviously reduced compared to the BG group. The above-mentioned metabolites were significantly modulated in SBS groups with different doses in comparison to the MG group. As the metabolic trend of the HD group was close to that of group BG, which means that there was no significant difference in the metabolites mentioned above, HD vs. BG is not shown in Table 2. These results suggested that SBS can regulate metabolic disorders through the key metabolites analyzed above. 

#### 3.6.3. Metabolic Pathway Enrichment Analysis

The Enrichment Analysis was conducted based on all the metabolites. We set all the substances with KEGG annotation as the background, without screening. The KEGG Enrichment formulation is as follows:$$P=1-\overset{m-1}{\underset{i=0}{\sum }}\frac{\left(\begin{array}{c} M\\ i \end{array}\right)\left(\begin{array}{c} N-M\\ n-i \end{array}\right)}{\left(\begin{array}{c} N\\ n \end{array}\right)}$$
where *N* denotes the number of metabolites with KEGG annotation in the entire list of metabolites, and *n* is the number of differential metabolites in *N* metabolites. In addition, *M* represents the number of metabolites of a KEGG pathway in *N*, and *m* is defined as the differential metabolite number of a KEGG pathway in *M* metabolites. The enrichment analysis process was carried out using our script, and the ggplot2 was used to draw the plots. The rich factor means *m*/*M*. 

Figure 9 shows the metabolic pathway enrichment analysis. Using MetaboAnalyst 3.5, the pathways affected by SBS were analyzed. The dots in Figure 9 reveal that: (i) the larger the dot size, the greater the number differential metabolites that were enriched in the pathway; (ii) the redder the dot, the more significant the enrichment of the pathway. As shown in Table 2 and Figure 9, the metabolism of citrate cycle (TCA cycle), biosynthesis of unsaturated fatty acids, arginine-proline metabolism, alanine, aspartate and glutamate metabolism, niacin and niacinamide metabolism, fat digestion, absorption and glycerophospholipid metabolism, and glycerolipid metabolism were the most affected metabolic pathways when SBS regulated the disordered metabolic levels induced by liver injury. These results indicated that SBS affects the metabolism of liver tissue by regulating the above metabolic pathways, thus reducing the degree of liver injury.

### 3.7. Identification and Selection of Differential Genes

#### 3.7.1. Differential Gene Volcano Map

Figure 10 shows the differentially expressed genes that were screened out under the condition of |log_2_FoldChange| = 1 and FDR < 0.05. The overall distribution of differentially expressed genes can be seen in the volcano map. Compared to the BG group, there were 959 genes expressed differently in group MG, of which 379 genes were down-regulated and 580 genes were up-regulated. On the other hand, comparing the MG group with the three SBS groups, we found that: (i) 229 genes were expressed differently in the LD group, with 65 down-regulated genes and 164 up-regulated genes; (ii) 42 out of 75 differentially expressed genes in the MD were down-regulated and the rest were up-regulated; (iii) among 404 differentially expressed genes in the HD group, there were 211 down-regulated genes and 193 up-regulated genes. Furthermore, comparing HD with BG, there were only four differentially expressed genes, which were meaningless differences. These results suggested that SBS can regulate gene expression disorders in rats.

#### 3.7.2. Expression of Key Genes

There were three overlapping differentially expressed genes among the six groups. These overlapping genes probably represented the related genes that alleviated liver injury using SBS. Figure 11 indicates that SBS was capable of down-regulating the expression levels of CYP1A1 and Noct, and up-regulating the expression of TUBB6 in the model group. The level of CYP1A1 in HD group was close to that of the BG group. As a result, SBS showed the ability to down-regulate the expression of NOCT, thereby reducing the hepatocyte steatosis induced by CCl_4_. In addition, SBS increased the TUBB6 expression, which may be related to the inhibition of inflammatory response by SBS. The reduction in CYP1A1 overexpression in group MG indicated that SBS can inhibit lipid peroxidation and reduce lipid peroxidation products, thus mitigating the CCl_4_-induced liver damage.

## 4. Discussion

After being treated with CCl_4_, liver cytochrome P-450 in rats is activated, and then the trichloromethyl free radical (·CCl_3_) is produced, which is covalently coupled with cell macromolecules, thus affecting cell functions. The produced CCl_3_ attack the phospholipid molecules on the liver cell membrane, causing a lipid peroxidation reaction, as well as destroying the integrity of the structure and function of the liver cell membrane. Meanwhile, the levels of SOD, MDA, and other related lipid peroxidation indices in liver tissues are altered, ultimately leading to impaired liver function [20]. Therefore, liver cells are encouraged to release various enzymes into the blood, including ALT, AST, and AKP, which are sensitive indicators of liver cell damage [21]. In addition to the above reactions, CCl_4_ also causes inflammation in the body, resulting in changes in inflammatory cytokines in the blood (IL-6 and TNF-α) and in liver tissues (COX-2 and PGE_2_) [22]. In terms of metabolomics, after CCl_4_ is absorbed by the body, the level of cytochrome P450 increases, leading to a series of lipid peroxidation reactions, resulting in disorder of the TCA circulation, and other related pathways are also affected [23]. The present work studied the effects of SBS on pathological changes of liver tissues and liver indices in rats with acute liver injury induced by CCl_4_. Our previous study reported that SBS can reduce ALT, AST, AKP, TNF-α, and IL-6 levels in the blood of rats. 

The current study found that SBS reduces CYP1A1 and, therefore, affects CCl_4_-induced lipid peroxidation. Furthermore, the levels of SOD, MDA, as well as other related lipid peroxidation indices in liver tissues are regulated. Then, the TCA cycle is modulated. Due to the fact that TCA cycle connects to other pathways, as shown in Figure 12, these pathways are modulated. The experimental results (Section 3.7.2) showed that SBS can also regulate Noct and TUBB6 in rats. By investigating existing studies, we found that ALT, AST, AKP, TNF-α, and IL-6, which were analyzed in our previous study, have connections to this present work in three aspects: (i) IL-6 is related to branched-chain amino acids (BCAA); (ii) ALT and AST are associated with alanine, aspartic acid, and glutamate metabolism; (iii) there are connections between TNF-α and TUBB6. Detailed explanations about these connections and corresponding conclusions will be given in the following sections.

A large number of studies have shown that the liver has a strong antioxidant defense mechanism [24], and there is a certain relationship between lipid oxidation and liver diseases [25]. SOD helps to remove free radicals, inhibit lipid peroxidation, and reduce lipid peroxidation products in the defense system of the body [26]. GSH-Px is an important type of enzyme that catalyzes hydrogen peroxide (H_2_O_2_), which can scavenge lipid peroxidase caused by reactive oxygen species (ROS), thus protecting the structural and functional integrity of cell membranes [27]. It should be mentioned that the main function of CAT is to remove lipid hydroperoxide and H_2_O_2_. After SOD disproportionates oxygen free radicals into H_2_O_2_, CAT decomposes them into water, thereby removing active oxygen in the body [28]. T-AOC is an indicator that reflects the state of the body’s antioxidant defense system. The higher activity of T-AOC and SOD in the body means a faster removal of free radicals [29]. The content of lipid peroxidation product malondialdehyde (MDA) reflects the degree of tissue peroxidation damage, and the damage of cells can also be indirectly reflected [30]. Furthermore, CCl_4_ increases the levels of PGE_2_, COX-2, and γ-GT, and decreases the TP level. The content of prostaglandins is closely related to the occurrence and development of inflammation [31]. In particular, PGE_2_ plays the main role in the inflammatory process and is an important mediator of the inflammatory response [32]. As a kind of inducible enzyme, COX-2 also participates in liver inflammatory injury. When cells are stimulated by pathogenic factors, the synthesis and release of COX-2 will be induced [33]. Moreover, γ-GT plays a critical role in the clinical diagnosis of liver injury diseases, which is also one of the most important indicators for evaluating liver functions [34]. In addition, the TP level reflects the synthetic ability of the liver [35]. In the present study, SBS was found to significantly increase the levels of antioxidant enzymes (SOD, GSH-Px, and CAT), and the activity of T-AOC (Section 3.2). In addition, the levels of γ-GT and PGE_2_, and the content of MDA, were reduced (Section 3.1). These results suggest that SBS has the effect of antagonizing acute liver injury, and the mechanism is probably related to the anti-lipid peroxidation and inhibition of the inflammatory reaction.

The TCA cycle is reported as the final common oxidation pathway of fat, carbohydrates, and amino acids, and is also the most important central pathway that connects almost all metabolic pathways of individuals [36]. In addition, the TCA cycle is closely related to some liver diseases. The accumulation of lipids and loss of insulin lead to two changes in the livers of mice: (i) an increase in oxidation and (ii) the replenishment pathways of the TCA cycle [37]. Slowing down the TCA cycle can reduce the production of reactive oxygen species (ROS) and protect the liver [38]. In this work, as shown in Table 2 and Figure 12, SBS reduces the level of malic acid in the MG and alleviates the disorder of the TCA cycle.

Arginine is known as a semi-essential amino acid, which has many biological functions in the human body [39]. It is also an intermediate in the urea cycle, the substrate for protein synthesis, and acts as the precursor of nitric oxide (NO), proline, glutamic acid, polyamines, and creatine [40]. In this study, as shown in Table 2 and Figure 12, after treatment with CCl_4_, there is an abnormal increase in the level of arginine caused by creatine, but the SBS treatment reduces the creatine level in liver tissues. It indicates that SBS has the ability to mediate the dysfunction of creatine or any arginine intermediate products that could be converted into creatine, which means that SBS can effectively reduce liver damage in rats. 

Branched-chain amino acids (BCAA), including leucine, isoleucine, and valine, play important roles in the liver, such as inducing mitochondrial biogenesis, inhibiting the production of reactive oxygen species (ROS), and stimulating both the production of hepatocyte growth factors and the synthesis of albumin and glycogen. BCAA has also been found to participate in hepatocyte apoptosis and improve insulin resistance [41]. In addition, branched-chain amino acids have been shown to: (i) effectively reverse CCl_4_-induced liver injury by down-regulating the TGF-β1 and Smad3/Smad7 signaling pathways; (ii) reduce the pro-fibrotic mediators TGF-β1 and IL-6; and (iii) improve glucose metabolism and insulin resistance [42]. Leucine is a kind of BCAA, which stimulates hepatic stellate cells (HSCs) to produce the hepatocyte growth factor for promoting protein synthesis. The simulation means that leucine also has a positive effect on patients with chronic liver disease [43]. In the present study (shown in Table 2 and Figure 12), compared to group BG, the leucine level in group MG decreased, while the level of leucine increased significantly when using SBS. In our previous study, SBS showed the effect of decreasing the level of IL-6. This result is consistent with the conclusion of this study, i.e., SBS can alleviate the liver injury in rats by increasing the level of BCAA and decreasing the level of IL-6.

Nicotinate is known as one of the 13 essential vitamins for the human body, which can be converted into nicotinamide [44]. Nicotinamide is a component of coenzyme I and II, and participates in lipid metabolism, tissue respiratory oxidation processes, and the anaerobic decomposition of carbohydrates [45]. Glycerin phospholipid is the most abundant phospholipid in the human body [46], known as an important bile component and membrane surfactant. It is also involved in the recognition of protein signal transduction through the cell membrane [47]. Alanine, aspartate, and glutamate metabolism, involved in the immune regulation system and in lymphocyte regeneration [48], are related to ALT and AST. By analyzing the results shown in Table 2 and Figure 12, SBS was found to alleviate CCl_4_-induced liver injury by improving the biological pathways that were mentioned above. 

Cytochrome P4501A1 (CYP1A1) is known as a subtype of the cytochrome P450 family, which was found to be an important regulator of reactive oxygen generation [49]. CYP1A1 is capable of inducing oxidative stress as a result of the over-production of reactive oxygen [50]. Some studies have shown that blocking CYP1A1 partially inhibits MDA deposition, indicating that the deletion of CYP1A1 helps to alleviate excessive deposition of lipid peroxide, and the concentration of ROS and SOD can be affected by the treatment of HepG2 cells with CYP1A1-siRNA [51]. This means that CYP1A1 regulates lipid peroxidation by affecting the ROS and SOD levels [52]. Furthermore, overexpression of CYP1A1 was found to increase the level of MDA and reduce the concentration of SOD, further indicating the importance of CYP1A1 in lipid peroxidation [53]. The present study shows that SBS reduces CYP1A1 overexpression in group MG (shown in Figure 11 and Figure 12). This means that SBS is capable of inhibiting lipid peroxidation and reducing lipid peroxidation products. These findings are consistent with the detection results of MAD and SOD in our experiments.

TUBB6 belongs to one type of β-tubulin. Cell proliferation is closely related to the synthesis of β-tubulin. The positive and negative effects of regulatory factors on β-tubulin are likely to be able to regulate cell proliferation [54]. Existing studies have shown that TNF-α not only inhibits cell proliferation, but also inhibits the expression of β-tubulin mRNA [55]. TNF-α is capable of: (i) down-regulating the number of parathyroid hormone receptors, (ii) inhibiting the concentration of cAMP and cytosolic free Ca^++^; and (iii) reducing the polymerized tubulin [56]. Our previous study found that SBS plays important roles in effectively inhibiting the inflammatory response of hepatocytes mediated by TNF-α, protecting hepatocytes, and significantly reducing the inflammatory lesions and structural damage of hepatocytes, thereby evidently improving liver functions of rats with acute liver injury. In this study, SBS increased TUBB6 in the MG group (Section 3.7.2), which was probably related to the decrease in the TNF-α level caused by SBS. The decreasing of the TNF-α level will reduce the inhibition of TUBB6. This mechanism alleviates the damage of hepatocytes caused by CCl_4_.

Nocturnin (NOCT) belongs to the superfamily of exonuclease, endonuclease, and phosphatase. Previous works have shown that NOCT null mice (*Noct* −/−) are resistant to high-fat-diet-induced obesity and hepatic steatosis. Further studies indicate that *Noct* −/− mice show a decrease in circulating triglyceride levels [57]. In 3T3-L1 cell lines, NOCT overexpression enhances adipogenesis when cells are cultured in adipogenic medium, whereas the reduction in NOCT in 3T3-L1 cells decreases their potential adipogenesis [58]. These results suggest that NOCT is essential for the regulation of lipid transport. In the current study, SBS showed the ability to down-regulate the expression of NOCT, thereby reducing the hepatocyte steatosis induced by CCl_4_.

This study provides the experimental basis for the liver-protective effects and mechanisms of SBS. The significant protective effects of pre-treated SBS on CCl_4_-induced liver injury provide solid support for our further study on the post-treatment effect of SBS. Referring to previous works [59], the pre-treated experiment is also a common way to study the protective effect of a substance that needs to be analyzed in rats. We believe that the present study is sufficient to draw a conclusion that SBS is endowed with protective effects against CCl_4_-induced liver injury in rats. We will consider the vitro and vivo experiments on the effect of SBS on CYP1A1 and Noct in future work. 

## 5. Conclusions

In conclusion, the present study found that SBS effectively alleviated the liver inflammation and hepatocyte pathological damage of rats with acute liver injury induced by CCl_4_. In addition, SBS was able to alleviate the metabolic disorders. Furthermore, the expression levels of CYP1A1, NOCT, and TUBB6 were regulated, indicating the effects of improving the antioxidant capacity of liver tissues, reducing the steatosis of hepatocytes, and inhibiting the inflammatory reaction of hepatocytes, respectively. 

## Figures and Tables

**Figure 1 molecules-27-02224-f001:**
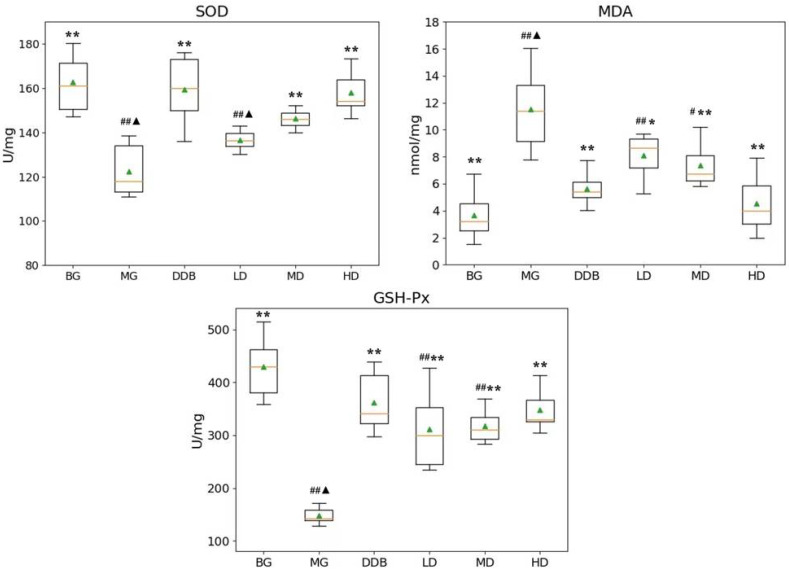
Effects of SBS on SOD, MDA, and GSH-Px in liver tissue. Data are presented as mean ± S.E.M. (*n* = 10). Significant differences compared with the BG group were designated as # *p* < 0.01 and ## *p* < 0.05, with the MG group as * *p* < 0.05 and ** *p* < 0.01, and with the DDB groups as ▲ *p* < 0.05.

**Figure 2 molecules-27-02224-f002:**
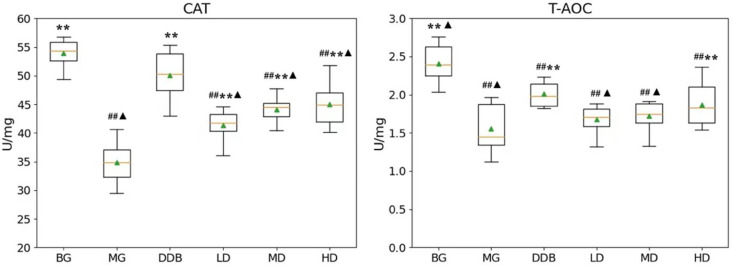
Effects of SBS on CAT and T-AOC in liver tissue. Data are presented as mean ± S.E.M. (*n* = 10). Significant differences compared with the BG group were designated as ## *p* < 0.05, with the MG group as ** *p* < 0.01, and with the DDB groups as ▲ *p* < 0.05.

**Figure 3 molecules-27-02224-f003:**
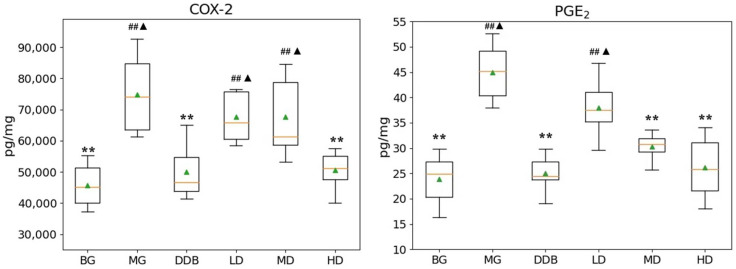
Effects of SBS on COX-2 and PGE2 in liver tissue. Data are presented as mean ± S.E.M. (*n* = 10). Significant differences compared with the BG group were designated as ## *p* < 0.05, with the MG group as ** *p* < 0.01, and with the DDB groups as ▲ *p* < 0.05.

**Figure 4 molecules-27-02224-f004:**
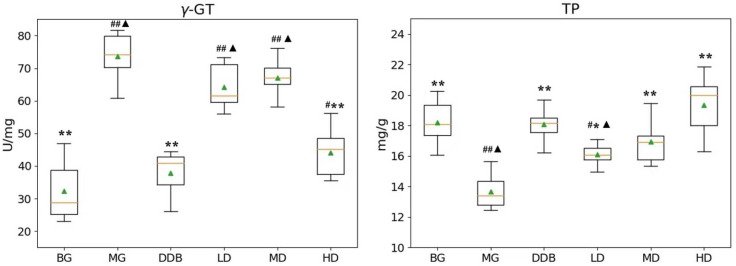
Effects of SBS on γ-GT and TP in liver tissue. Data are presented as mean ± S.E.M. (*n* = 10). Significant differences compared with the BG group were designated as # *p* < 0.01 and ## *p* < 0.05, with the MG group as * *p* < 0.05 and ** *p* < 0.01, and with the DDB groups as ▲ *p* < 0.05.

**Figure 5 molecules-27-02224-f005:**
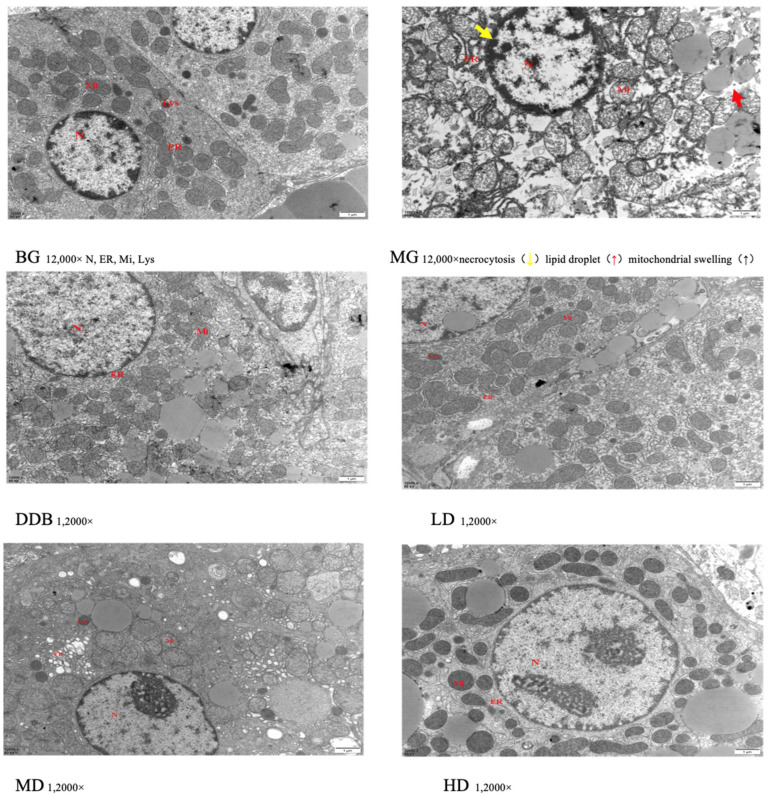
Electron microscopic observation.

**Figure 6 molecules-27-02224-f006:**
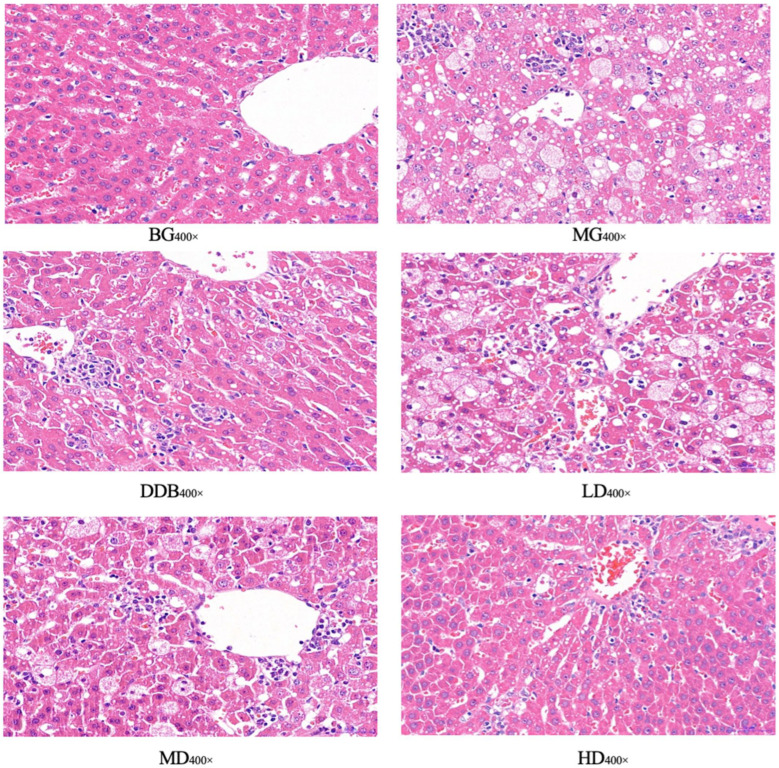
Light microscopic observation.

**Figure 7 molecules-27-02224-f007:**
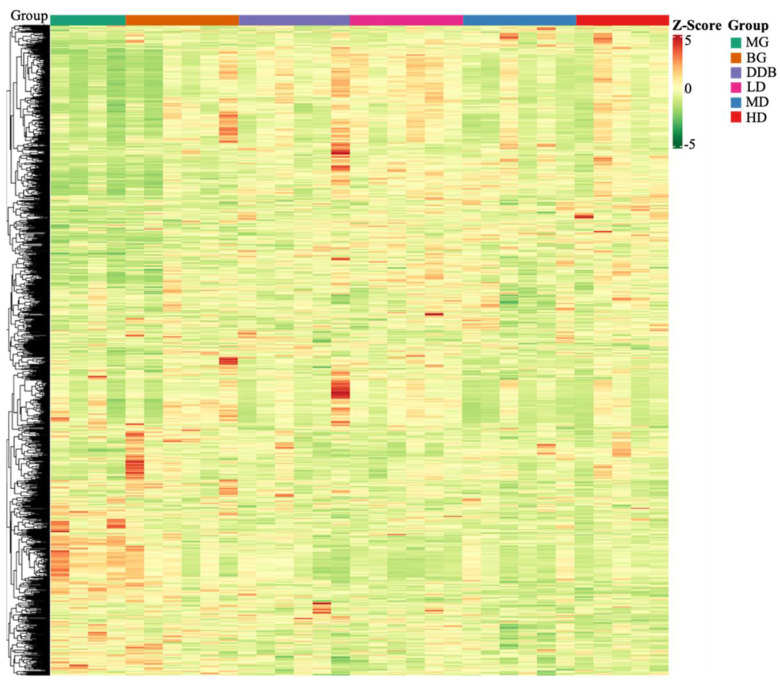
Heat map. The x-axis represents groups, and the y-axis represents differentially expressed genes. Data (peak areas) were normalized between −5 and 5 (green—the lowest level; red—the highest level).

**Figure 8 molecules-27-02224-f008:**
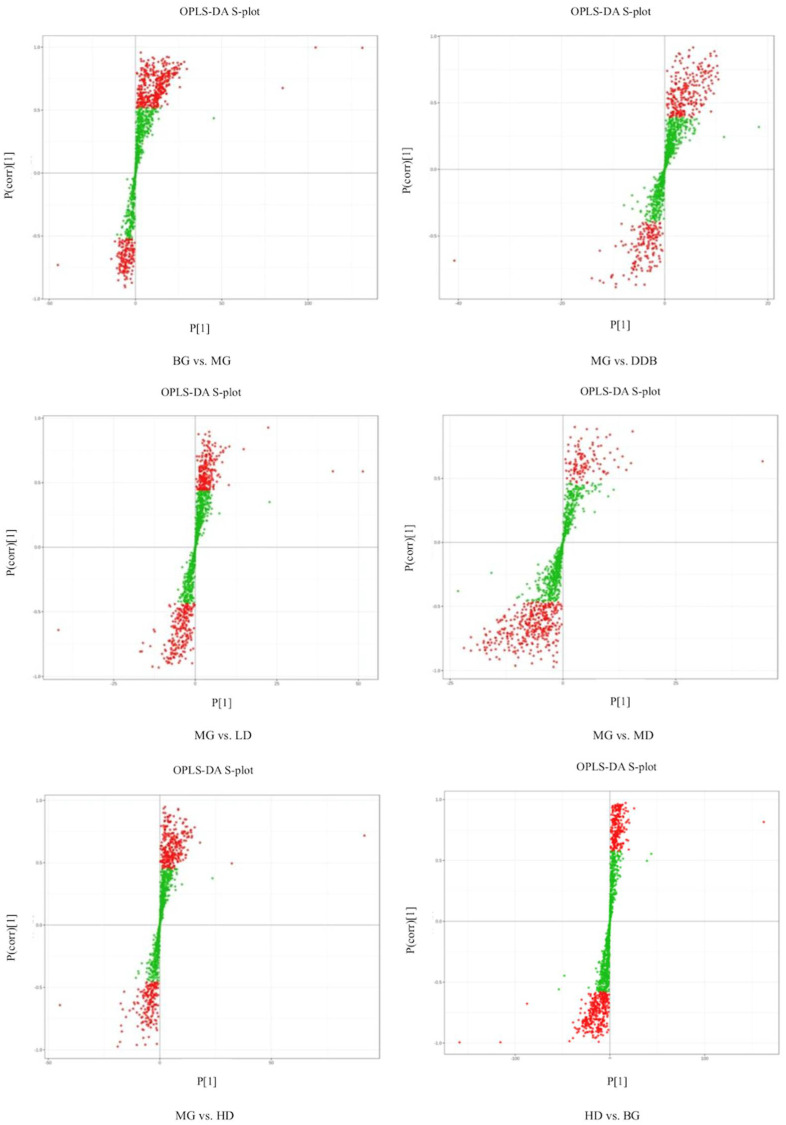
OPLS-DA S-plot. S-plot from OPLS-DA. Covariance and correlation were plotted on the x- and y-axis, respectively. The variables located far from the origin represent influence with high reliability in the model.

**Figure 9 molecules-27-02224-f009:**
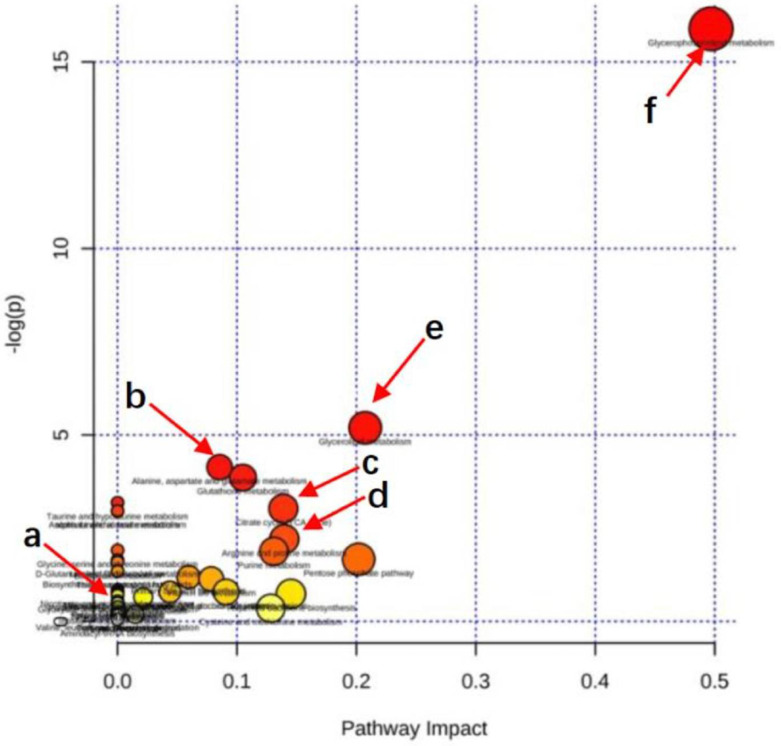
Pathway enrichment. Summary of pathway analysis of significantly reversed metabolites between rats in MG and SBS-treated rats. Rich factor and KEGG pathway were plotted on the x- and y-axis, respectively. The larger the dot, the greater the number of metabolites that were enriched in the pathway. The darker the dot color, the more significant the enrichment. (a) Nicotinate and nicotinamide metabolism. (b) Alanine, aspartate, and glutamate metabolism. (c) Citrate cycle (TCA cycle). (d) Arginine-proline metabolism. (e) Glycerophospholipid metabolism. (f) Glycerolipid metabolism.

**Figure 10 molecules-27-02224-f010:**
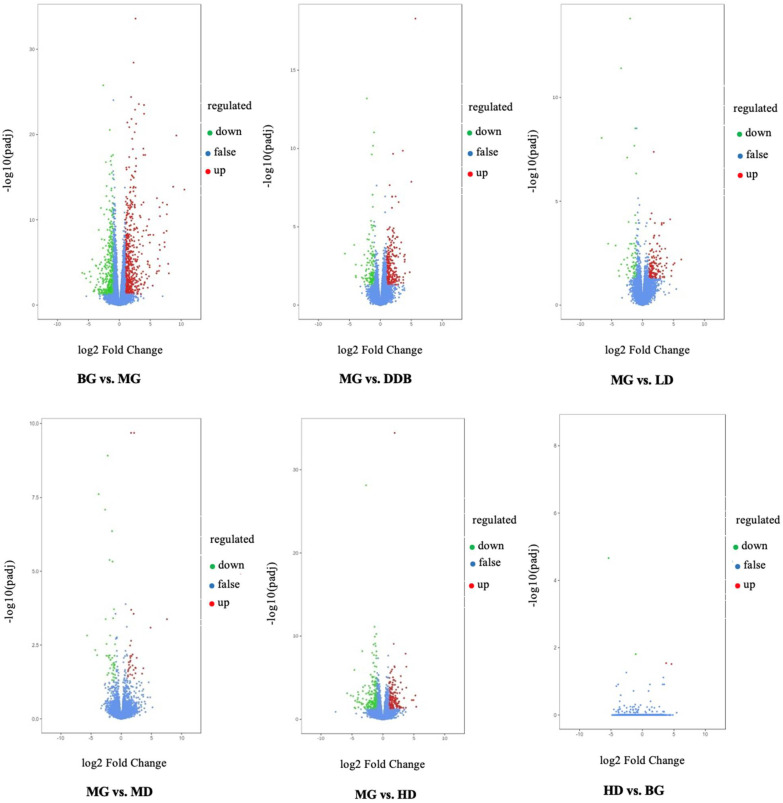
Volcano. Fold change in gene expression and significance level of the differential gene are plotted on the x- and y-axis, respectively. Up-regulated genes are represented by red dots, down-regulated genes are represented by green dots, and genes without differential expression are represented by blue dots.

**Figure 11 molecules-27-02224-f011:**
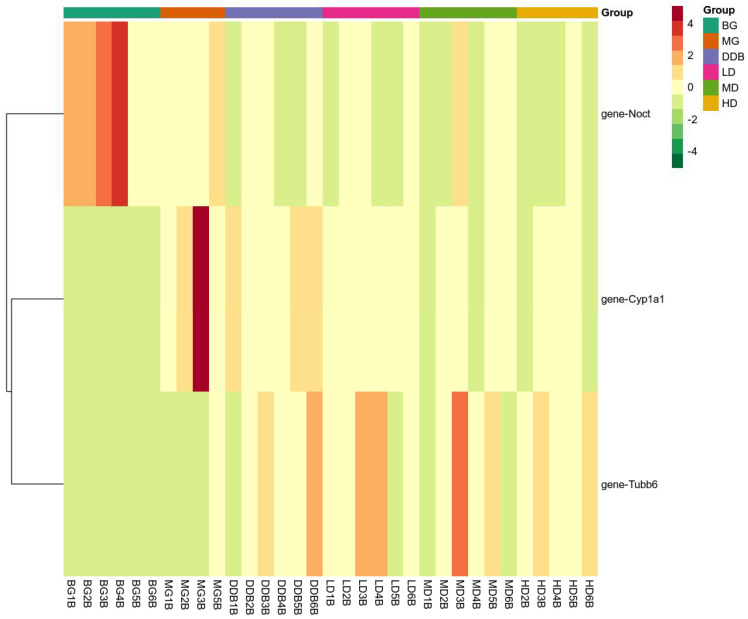
Heat map. The x-axis represents groups, and the y-axis represents differentially expressed genes. Data (peak areas) were normalized between −4 and 4 (green—the lowest level; red—the highest level).

**Figure 12 molecules-27-02224-f012:**
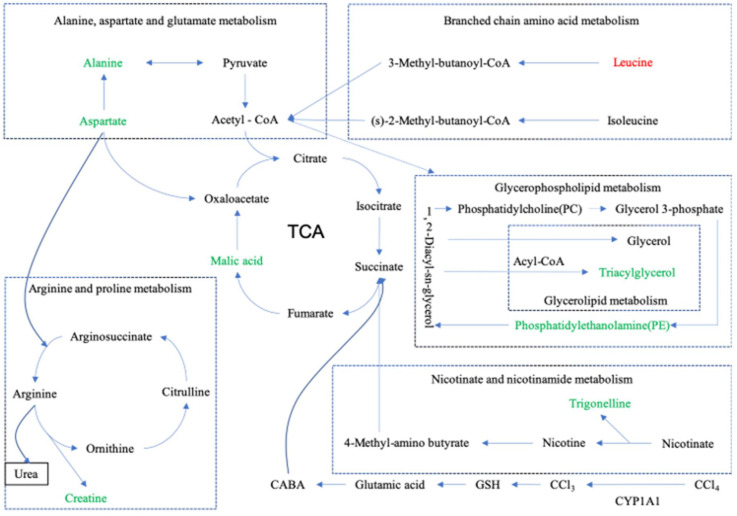
Metabolite pathway. Summary diagram of metabolic pathway relationships. The up-regulated substances are represented in red font, and the down-regulated substances are represented in green font. Dashed lines represent indirect relationships between substances, and solid lines represent direct relationships between substances.

**Table 1 molecules-27-02224-t001:** Pathological results.

Group	Results
BG	(−)
MG	Feathery degeneration of hepatocyte (++) Fatty degeneration of hepatocytes (++) Hepatocyte necrosis (++) Inflammatory cell infiltration (+)
DDB	A small amount of feathery degeneration of hepatocytes (+) A small amount of fatty degeneration of hepatocytes (+) Hepatocyte necrosis (+)
LD	Feathery degeneration of hepatocytes (++) Fatty degeneration of hepatocytes (+) Hepatocyte necrosis (+) Inflammatory cell infiltration (+)
MD	Feathery degeneration of hepatocytes (++) Fatty degeneration of hepatocytes (+) Hepatocyte necrosis (++) Inflammatory cell infiltration (+)
HD	A small amount of feathery degeneration of hepatocytes (+) A small amount of fatty degeneration of hepatocytes (+) Hepatocyte necrosis (+)

**Table 2 molecules-27-02224-t002:** Effects of SBS on metabolic in liver tissues of rats with acute liver injury induced by CCl_4_ (X ± s, *n* = 6).

NO.	Metabolites	RT	MG/BG	LD/MG	MD/MG	HD/MG	DDB/MG	Pathway
1	L-Malic Acid	0.88	↑ *	↓/-	↓ *	↓ *	↓/-	Citrate cycle (TCA cycle)
2	7Z, 10Z, 13Z, 16Z, 19Z-docosapentaenoic acid	11.3	↑ *	↓ *	↓ *	↓ *	↓/-	Biosynthesis of unsaturated fatty acids
3	creatine	0.78	↑ *	↓/-	↓ *	↓ *	↓ *	Arginine-proline metabolism
4	n-acetyl-l-alanine	1.3	↑ *	↓/-	↓ *	↓ *	↓ *	-
5	N-Acetylaspartate	0.69	↑ *	↓/-	↓ *	↓ *	↓ *	Alanine, aspartate and glutamate metabolism
6	Trigonelline	0.77	↑ *	↓/-	↓ *	↓ *	↓/-	Nicotinate and nicotinamide metabolism
7	4-guanidinobutyric acid	0.85	↑ *	↓ *	↓ *	↓ *	↓ *	Arginine-proline metabolism
8	N-Amidino-L-Aspartate	0.96	↑ *	↓ *	↓ *	↓ *	↓ *	-
9	n-glycyl-l-leucine	1.8	↓ *	↑/-	↑ *	↑ *	↑/-	-
10	FFA(6:0)	0.73	↓ *	↑ *	↑ *	↑ *	↑ *	Fat digestion and absorption
11	CE(16:1)	13.12	↑ *	↓ *	↓ *	↓ *	↓ *	-
12	CE(18:2)	13.21	↑ *	↓ *	↓ *	↓ *	↓ *	-
13	PE(16:1/16:0)	6.25	↑ *	↓ *	↓ *	↓ *	↓ *	Glycerophospholipid metabolism
14	DG(16:0/20:2/0:0)	9.18	↑ *	↓/-	↓ *	↓ *	↓/-	Glycerolipid metabolism
15	TG(14:0/18:0/18:2)	11.85	↑ *	↓/-	↓ *	↓ *	↓ *	Glycerolipid metabolism
16	TG(14:0/18:0/20:4)	11.81	↑ *	↓/-	↓ *	↓ *	↓ *	Glycerolipid metabolism
17	TG(16:0/16:1/22:5)	11.41	↑ *	↓/-	↓ *	↓ *	↓ *	Glycerolipid metabolism

* *p* < 0.05, there was a significant difference between two groups. /-, there was no significant difference between two groups.

## Data Availability

All data generated and analyzed in this paper have been uploaded to the Figshare website: 10.6084/m9.figshare.18099515 (accessed on 1 March 2022), 10.6084/m9.figshare.19158421 (accessed on 1 March 2022), 10.6084/m9.figshare.18142718 (accessed on 1 March 2022).

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
