# Peer review of "Metabolomic Profiling Reveals Protective Effects and Mechanisms of Sea Buckthorn Sterol against Carbon Tetrachloride-Induced Acute Liver Injury in Rats"

_molecules, 2022, doi:10.3390/molecules27072224_

Round 1
Reviewer 1 Report
The paper is investigating the positive effects of sea buckthorn sterols on acute liver injury. The authors used numerous techniques to uncover the effects of buckthorn sterols on liver functions. They have found convincing evidences on the positive effects and build up model (working hypothesis) how it could be effective. The paper is well written.
However data presentation and calculations are raising some questions, moreover in some cases it is hard to understand the used expressions.
Please let me give an overview about these concerns.
Conceptually it is hard to understand why CCl4 induced liver injury model was used, there are many (sometimes more realistic /relevant) models. I would happy to see some justification about this in the manuscript.
In the abstract it is hard to understand what it means: „levels of metabolic pathways”
Metabolomics: It was not described accurately how metabolites were identified and quantified (eg: retention time deviance from standards, calibration curves were used or not, normalization, etc)
Statistical analysis on metabolomics data: It is not described that the authors used or not FDR corrections, I would strongly recommend to use it in case of metabolomics also not only in transcriptomics.
It is hard to understand what „metabolic trends” means:
„In addition, metabolic trends of LD, MD and HD groups were highly different from that of the MG group. Meanwhile, the metabolic trend of HD group was close to that of group BG”
From PLS-DA it is not possible to conclude proximity between sample groups because PLS-DA is forcing to find and overweight differences between sample groups. I would strongly recommend using standardized Euclidean distance to support claims like this. For representation using hierarchical clustering is recommended.
I think using „Determination of Relevant Biomarkers” the biomarker is not the correct expression here, diseases have biomarkers. These are metabolite level differences.
„And the t-test was employed to test the significant difference of metabolites between the groups.”
Could you please support your findings with using correction to multiple testing or if you have used please describe in the manuscript.
„Referring to literature reports and searching in the online database (HMDB), 17 metabolites (shown in Table 2)”
Could you please describe more accurately how have you selected that given metabolites, which were the selection criteria, that would improve the paper.
Please let me ask that: have you used custom background metabolit set during Metabolic Pathway Enrichment Analysis. Using background metabolit set (what you reliably quantified) improves the reliability of enrichment calculations. Which type of enrichment calculations was performed, the compound list based or the concentration table based. Could you please specify this important methodological detail in manuscript.
Plots:
I would strongly recommend to use boxplots for data representation where it is possible. Figure 1;2;3;4 is not acceptable in this way of representation.
In the link, which should contain all the supplementary data, the metabolomics data is not available (just the transcriptomics). I would happy if you would share you data with the scientific community upon acceptance.
Author Response
The paper is investigating the positive effects of sea buckthorn sterols on acute liver injury. The authors used numerous techniques to uncover the effects of buckthorn sterols on liver functions. They have found convincing evidences on the positive effects and build up model (working hypothesis) how it could be effective. The paper is well written.
However data presentation and calculations are raising some questions, moreover in some cases it is hard to understand the used expressions.
Please let me give an overview about these concerns.
Conceptually it is hard to understand why CCl4 induced liver injury model was used, there are many (sometimes more realistic /relevant) models. I would happy to see some justification about this in the manuscript.
Thanks a lot for your comments.
CCl4 is commonly used in a mouse model of live injury, which is responsible for oxidative stress and lipid peroxidation through the cytochrome P450-mediated generation of the highly reactive CCl3·, leading to eventual cellular damage characterized by hepatocellular necrosis [1]. The biochemical and histological events of correspinding damage induced by CCl4are quite similar to the liver cirrhosis in humans. Through oral or gavage feedings, reproducible and obvious acute liver injury can be induced, helping to study pathogenic mechanisms or protection mechanisms related to the liver. We have added the justification in the latest manuscript.
In the abstract it is hard to understand what it means: „levels of metabolic pathways”
Thanks for pointing this out.
We use the term of ‘level’ in the abstract to represent the mening ---‘expression level of metabolite in metabolic pathways’. Once the expression level of metabolite changes, the entire metabolic pathway will change. We agree that the term ‘level’ is infelicitous, so we use ‘expression’ to replace it in the manuscript.
Metabolomics: It was not described accurately how metabolites were identified and quantified (eg: retention time deviance from standards, calibration curves were used or not, normalization, etc)
The corresponding description has been added in the manuscript.
Based on the self-built Metware Database (MWDB), qualitative analysis was performed according to the retention time (RT), ion-pair information, and secondary spectrum data. The retention time of metabolite has been given in Table 2.
Statistical analysis on metabolomics data: It is not described that the authors used or not FDR corrections, I would strongly recommend to use it in case of metabolomics also not only in transcriptomics.
FDR corrections were used in metabolomics. The corresponding revision has been made in the manuscript.
It is hard to understand what „metabolic trends” means:
The expression level of metabolite is different among different groups, leading to variant global metabolic statuses in different groups, i.e., each group has its own expression status of metabolite. To appropriately express this meaning, we use ‘global metabolic status’ to replace the ‘metabolic trends’ in the manuscript.
“In addition, metabolic trends of LD, MD and HD groups were highly different from that of the MG group. Meanwhile, the metabolic trend of HD group was close to that of group BG”
From PLS-DA it is not possible to conclude proximity between sample groups because PLS-DA is forcing to find and overweight differences between sample groups. I would strongly recommend using standardized Euclidean distance to support claims like this. For representation using hierarchical clustering is recommended.
Thanks for this constructive suggestion to improve our manuscript.
Thehierarchicalclustering map has been added tothe new manuscript for better representation.
I think using „Determination of Relevant Biomarkers” the biomarker is not the correct expression here, diseases have biomarkers. These are metabolite level differences.
Many thanks for your suggestion. We have modified the ‘Determination of Relevant Biomarkers’ torelevantmetabolite.
„And the t-test was employed to test the significant difference of metabolites between the groups.”
Could you please support your findings with using correction to multiple testing or if you have used please describe in the manuscript.
In addition to t-test, ANOVA is also used to analyze the differences between metabolites. The corresponding content has been revised.
„Referring to literature reports and searching in the online database (HMDB), 17 metabolites (shown in Table 2)”
Could you please describe more accurately how have you selected that given metabolites, which were the selection criteria, that would improve the paper.
The criteria for selecting the metabolite is to select the common differential metabolites that appeare in each group, which are related to acute liver injury. Through this selection criteria, we can compare differences among groups, and further analyze the protection effects of sea buckthorn sterolon acute liver injury.
We havedescribed this in the manuscript.
Please let me ask that: have you used custom background metabolit set during Metabolic Pathway Enrichment Analysis. Using background metabolit set (what you reliably quantified) improves the reliability of enrichment calculations. Which type of enrichment calculations was performed, the compound list based or the concentration table based. Could you please specify this important methodological detail in manuscript.
The Enrichment Analysis is calculated based on all the metabolites. We set all the substances with KEGG annotation as the backgroud, without screening.
The KEGG Enrichment formulation is as follows:
where N denotes the number of metabolites with KEGG annotation in the entire list of metabolites, and n is the number of differential metabolites in N metabolites. In addition, M represents the number of metabolites of a KEGG pathway in N, and m is defined as the differential metabolite number of a KEGG pathway in M metabolites. The enrichment analysis process is carried out by our script, and the ggplot2 is used to draw plots. The rich factor means m/M.
The corresponding explanation has been added in the manuscript.
Plots:
I would strongly recommend to use boxplots for data representation where it is possible. Figure 1;2;3;4 is not acceptable in this way of representation.
Thanks for your helpful suggestions. We now use the boxplots for representation in the revised manuscript.
In the link, which should contain all the supplementary data, the metabolomics data is not available (just the transcriptomics). I would happy if you would share you data with the scientific community upon acceptance.
The metabolomics data has been deposited in the public curated repository Figshare. Link: 10.6084/m9.figshare.18099515
[1] Paulpriya K, Tresina P S, Mohan V R. Hepatoprotective effect of crotalaria longipes wight and arn, ethanol extract in CCL4 induced hepatotoxicity in wistar rats[J]. Int. J. Toxicol. Pharmacol. Res, 2016, 8: 45-52.

Reviewer 2 Report
This article investigated the protective effect of sea buckthorn sterol (SBS) against carbon tetrachloride (CCl4)-induced acute liver injury in rats and its mechanism. The experimental results showed that SBS has a protective effect on CCl4-induced liver injury in rats, and the analysis of its mechanism may be related to the regulation of metabolic disorders, anti-lipid peroxidation, and inhibition of inflammatory response. A few comments are as follows.
- In the introduction part of the “Sulfur dioxide and Carbon tetrachloride” please standardize the writing.
- There should be a space between the numbers and units in the 1 section-“0.5mg/100g” and 2.6 section –“2.6μm” and “60/40V 10/90V”.
- In addition, the form of a liter in the article should be uniform, not both “l” and “L”.
- What is the cause of the large standard deviation in Figure 1?
- And with such a large deviation why the difference between the two groups is not significant?
- It is suggested to add some comparative literature with sterols in other plants in the discussion section.
The topic of this paper has some practical value and the experimental data is sufficient, there are some mistakes in format and writing, I hope to correct them.
Author Response
This article investigated the protective effect of sea buckthorn sterol (SBS) against carbon tetrachloride (CCl4)-induced acute liver injury in rats and its mechanism. The experimental results showed that SBS has a protective effect on CCl4-induced liver injury in rats, and the analysis of its mechanism may be related to the regulation of metabolic disorders, anti-lipid peroxidation, and inhibition of inflammatory response. A few comments are as follows.
- In the introduction part of the “Sulfur dioxide and Carbon tetrachloride” please standardize the writing.
Thanks a lot for your comments and pointing this out. The corresponding revision has been made.
- There should be a space between the numbers and units in the 1 section-“0.5mg/100g” and2.6 section –“2.6μm” and “60/40V 10/90V”.
The manuscript has been revised as suggested by you.
- In addition, the form of a liter in the article should be uniform, not both “l” and “L”.
Thanks for your careful reading. It has been revised.
- What is the cause of the large standard deviation in Figure 1?
The large standard deviation shows due to the calculation error.We have modified this. This finding is helpful and appreciated.
- And with such a large deviation why the difference between the two groups is not significant?
As mentioned in the last comment, the mistake has been corrected. The latest results can be found in the boxplots in the revised manuscript.
- It is suggested to add some comparative literature with sterols in other plants in the discussion section.
This suggestion is useful to improve our manuscript. Thanks again. We have carefully selected some literature and added the description in the discussion section.
-sitosterol, a naturally occurring sterol molecule, is a relatively mild to moderate antioxidant and exerts beneficial effects in vitro by decreasing the level of reactive oxygen species. The previous study [2] has evaluated the antioxidant potential of -sitosterol in 1,2-dimethylhydrazine (DMH)-induced colon carcinogenesis. There has been other study which suggests that phytosterol esters have [3] effects on countering hypercholesterolemia-related changes in the brain by decreasing the cholesterol levels, increasing the phospholipid levels and increasing the level of antioxidant enzymes.
[2] Baskar A A, Al Numair K S, Gabriel Paulraj M, et al. β-sitosterol prevents lipid peroxidation and improves antioxidant status and histoarchitecture in rats with 1, 2-dimethylhydrazine-induced colon cancer[J]. Journal of Medicinal Food, 2012, 15(4): 335-343.
[3] Song L, Zhou H, Yu W, et al. Effects of phytosterol ester on the fatty acid profiles in rats with nonalcoholic fatty liver disease[J]. Journal of Medicinal Food, 2020, 23(2): 161-172.

Reviewer 3 Report
- It is suggested to change the abbreviation of section 2.2, specifically DBB for DDB, since the latter appears in all the figures in this article.
- It is important to justify the low, medium and high doses through a calibration curve.
- In the figures that contains the photos, include the calibration bars inside each one.

Author Response
The topic of this paper has some practical value and the experimental data is sufficient, there are some mistakes in format and writing, I hope to correct them.
- It is suggested to change the abbreviation of section 2.2, specifically DBBfor DDB, since the latter appears in all the figures in this article.
Thanks a lot for your comments and suggestions. The corresponding revision has been made.
- It is important to justify the low, medium and high doses through a calibration curve.
The selection basis of the low, medium and high doses is set according to existing literature, which has been cited in the manuscript.
- In the figures that contains the photos, include the calibration bars inside each one.
This suggestion is quite helpful for the clear demonstration of some experimental results. In our manuscript, Figures 5 and 6display the observation photos of different groups under the same amplification times of light microscope and electron microscope, respectively. The amplification times have been annotated in the figures. As the amplification time of photos in a figure is the same, we can directly analyze and compare the pathological changes.

Round 2
Reviewer 1 Report
Dear Authors,
I am grateful for your answers and clarifications. I accept them.
However, I strongly recommend to improve the resolution of the figures.